# Signal and Nutritional Effects of Mixed Diets on Reproduction of a Predatory Ladybird, *Cheilomenes propinqua*

**DOI:** 10.3390/insects14070587

**Published:** 2023-06-28

**Authors:** Andrey N. Ovchinnikov, Antonina A. Ovchinnikova, Sergey Y. Reznik, Natalia A. Belyakova

**Affiliations:** 1Zoological Institute, Russian Academy of Sciences, Universitetskaya 1, 199034 St. Petersburg, Russia; anovchi@gmail.com (A.N.O.); antoninaovch@gmail.com (A.A.O.); 2All-Russia Institute of Plant Protection, Russian Academy of Sciences, Podbelskogo 3, Pushkin, 196608 St. Petersburg, Russia; belyakovana@yandex.ru

**Keywords:** reproduction, oogenesis, fecundity, resorption, diapause, food, biological control, *Cheilomenes propinqua*, *Sitotroga cerealella*, *Myzus persicae*

## Abstract

**Simple Summary:**

Biological control of insect pests is the main alternative to the extensive application of chemicals. Mass rearing of biocontrol agents requires the development of a cost-effective diet. We evaluated the potential of using a mixed diet consisting of one low-quality (the grain moth eggs) and one high-quality (the green peach aphid) prey species as food for females of a predatory ladybird *Cheilomenes propinqua*. The fecundity of females fed only on the grain moth eggs was very low. Daily consumption of two aphids increased the proportion of egg-laying females and daily consumption of 10 aphids resulted in an increase in their mean fecundity. Thus, the use of a mixed diet can be considered a promising technique for mass rearing of *C. propinqua,* although the economic feasibility of this method would most probably require the improvement of aphid rearing system. The fecundity of *C. propinqua* females used for biological control of pests in greenhouses by preventing colonization and supplied with the grain moth eggs will be low but the appearance of pests will cause a proportional increase in the mean fecundity of ladybirds.

**Abstract:**

It is known that food has a double impact on females of predatory ladybirds: qualitative signal effect (the onset of oogenesis) and quantitative nutritional effect (the increase in oogenesis intensity). We compared the patterns of these effects by feeding *Cheilomenes propinqua* females on mixed diets: unlimited low-quality prey (eggs of the grain moth *Sitotroga cerealella*) and limited high-quality prey (the green peach aphid *Myzus persicae*: 0, 2, 10, and 50 aphids per day). About half of the females fed only on the grain moth eggs oviposited and their fecundity was very low. Daily consumption of 2 aphids increased the proportion of egg-laying females whereas only consumption of 10 aphids increased their mean fecundity. Thus, the threshold of the signal effect was lower than that of the nutritional effect. As applied to mass rearing, we conclude that the addition of high-quality prey to low-quality food causes a substantial increase in egg production, although the economic feasibility of this method is not clear. Regarding biological control of pests by preventing colonization, we conclude that the fecundity of *C. propinqua* females supplied with the grain moth eggs in the absence of aphids will be low but the appearance of pests will cause a proportional increase in the mean fecundity of ladybirds.

## 1. Introduction

Food is one of the most important environmental factors influencing insect physiology and behavior. All insects show more or less high food specificity. The problem of food specificity is particularly important for the mass rearing of biocontrol agents where usually a compromise is required between the cheapness of production and the insect preference for various foods. The right choice of proper diet ensures a combination of low cost and high quality, and hence economic efficiency of mass rearing. Therefore, detailed knowledge of feeding specificity is a necessary prerequisite for a successful biocontrol project [1,2,3,4,5].

Acceptable foods of predatory ladybirds can be separated into two types: (1) ‘essential’ food that ensures oogenesis, stimulates oviposition, and supports normal larval growth and development, and (2) ‘alternative’ food that can be used only as a source of energy ensuring the survival of adults [6,7]. Most often, essential foods are aphids, coccids, mites, thrips, and other small soft-bodied prey whereas alternative foods are nectar and other substances of plant origin. Essential foods in their turn can be separated into high- and low-quality ones differing markedly in both preference and performance. In most cases, rankings based on different parameters (feeding preference, larval growth and development, female maturation and fecundity, etc.) coincide [6,7,8,9,10] although exceptions to this rule were also recorded [11,12].

Moreover, feeding some coccinellids on alternative or low-quality essential food results not only in the delay or termination of oogenesis but also in the decrease of metabolism, the accumulation of fat, and other components of the adult (reproductive) diapause. This so-called ‘trophic diapause’ has been demonstrated not only in predatory [13,14,15,16,17,18,19,20] but also in phytophagous ladybirds [21,22]. Thus, food can have a double impact on the reproduction of coccinellids: (1) ‘signal’ effect (as an environmental diapause-inducing or diapause-averting cue) and (2) ‘nutritional’ effect (specific nutrients are necessary for oogenesis).

The influence of diet on female maturation and fecundity of predatory ladybirds is particularly important for the development of a cost-effective mass-rearing method. Although in laboratory studies the highest preference and performance are usually observed with feeding on natural hosts, this food is usually too expensive for industrial mass rearing. This contradiction can be resolved by using various factitious foods. Rather often the optimal diet is a mixture of two or several complementary foods [6,7,8,12,18,23,24,25,26,27,28,29,30,31,32,33,34,35,36,37,38,39,40,41]. The complexity of the mixed foods studied varied widely from wholly artificial diets including numerous natural and synthetic components e.g., [12,24,31,35,39] to simple mixtures of two prey species e.g., [15,18,29,33]. In some studies, the mixed diet consisted of low-quality food and a small proportion of high-quality prey e.g., [15,18,42]. In some cases, a substantial super-additive effect was observed: feeding on mixed diets gave better results than feeding on each of their components [12,18,23,28,32,34,36,37,39,40].

The present study was conducted on *Cheilomenes propinqua* Mulsant (Coleoptera: Coccinellidae). The natural geographic range of this ladybird includes Africa and the Middle East [43]. *Cheilomenes propinqua* is known as an active predator of aphids, soft scales, whiteflies, mealybugs, and other small sucking pests. This predatory ladybird shows not only a wide prey range, but also broad habitat specificity reaching high population densities in fields, orchards, vineyards, and other agricultural landscapes [44,45,46,47,48,49,50,51,52,53,54]. Preliminary laboratory studies showed that *C. propinqua* has a number of desirable properties of a potential successful mass-reared biocontrol agent: rapid pre-adult development (10–11 days), high voracity (depending on prey species, larva consumed a total of about 500 aphids, female consumed 40–80 aphids per day), high lifetime fecundity (about 1000 eggs per female) and prey search activity [20,55,56,57,58,59]. Indeed, a recent study showed that *C. propinqua* can be successfully used for the biological control of greenhouse pests [56].

However, wide-scale use of a predatory insect for biocontrol in greenhouses is impossible without the development of optimal methods for its mass rearing and storage, which in turn requires detailed knowledge of its ecophysiology. Earlier laboratory studies conducted on *C. propinqua* concerned the factors influencing the rate of development and maturation, fecundity, walking activity, and storage potential [20,44,45,55,58,59]. In these studies, various foods were tested but the problem of selection of a cost-effective diet for laboratory and mass rearing still remains unsolved.

The main aim of the present study was to evaluate the potential of using a mixed diet consisting of one low-quality (eggs of the Angoumois grain moth *Sitotroga cerealella* Oliv) and one high-quality (the green peach aphid *Myzus persicae* (Sulz.)) prey species for *C. propinqua* females. The selection of just such a combination of prey species was based on our previous studies [20,58]. Feeding on the green peach aphid ensured rapid pre-adult development, low larval mortality, fast maturation, and high fecundity of *C. propinqua* females. Evidently, *M. persicae* (as well as some other aphids [44,45,55]) is a high-quality prey for this ladybird species. However, rearing aphids is a time- and labor-consuming (and hence rather expensive) process which is hardly amenable to automation. Rearing of the grain moth, on the contrary, can be easily automated and therefore it is a relatively cheap process. On the other hand, the grain moth eggs are a low-quality food that is practically not suitable for *C. propinqua* pre-adult development, and being used as female food results in slow maturation and low fecundity. Moreover, feeding on the grain moth eggs often causes the induction of reproductive diapause [20] suggesting that this food has not only nutritional but also signal negative effects on *C. propinqua* reproduction. If this is the case, an addition of even a small number of aphids would possibly prevent the induction of reproductive diapause and thereby would substantially increase the mean *C. propinqua* population fecundity.

Regarding the practical application of our study, it should be also noted that *C. propinqua* adults could be released for biological control of aphids in greenhouses by preventing colonization (a so-called ‘standing army’ approach). The application of this method often involves regular supplementation of factitious food to ensure the survival of biocontrol agents in the absence of pests [4,60,61,62,63,64,65,66,67,68]. The grain moth eggs could be used for this supplementation and then *C. propinqua* females would feed on eggs with the addition of some amount of aphids.

Our study also addresses fundamental aspects of insect eco-physiology. As noted above, food can have a double impact on ladybird reproduction. Reproductive maturation of ladybirds (onset of oogenesis) is at least partly triggered by the signal effect, whereas fecundity (oogenesis intensity) is mostly or even exclusively determined by the nutritional effect. Various trophic effects on the reproductive activity of predatory coccinellids have been intensively investigated [6,7,8,23]. However, in relatively few studies the effects of the quantity of a diet component on maturation (pre-oviposition period) and fecundity of ovipositing females were separated and the patterns of these two trophic responses were compared [15,18,39,69,70,71,72]. Thus, the fundamental aim of our study was to determine and compare the patterns of signal and nutritional effects of the addition of various quantities of high-quality prey to low-quality food-based diets.

## 2. Materials and Methods

### 2.1. Insects

The study was conducted on a laboratory population of *C. propinqua* originated from adults collected in 2015 in Alexandria, Egypt (31.200391° N, 29.9155046° E) and reared in All-Russian Institute of Plant Protection on the wheat aphid (*Schizaphis graminum* Rond.) at a temperature of about 24 °C and photoperiod of L:D = 16:8 (hereafter, light and dark periods in h are given). Before the experiment, ladybirds were reared for about one year in Zoological Institute RAS at a temperature of 25 °C and L:D = 16:8 feeding on the green peach aphid *M. persicae* which was reared on *Vicia faba* L. seedlings.

### 2.2. Experimental Design

The experiment was conducted at photothermal conditions that were shown to be close to optimal for the reproduction of this species [20]. At the beginning of each replicate of the experiment, a group of the first instar larvae hatched during 24 h from eggs laid by 10–15 *C. propinqua* females was reared feeding on the green peach aphid (hereafter also referred to as ‘aphids’) at 25 °C and L:D = 14:10. The emerged males and females were separated and then for 3 days males fed on aphids whereas females fed on the eggs of the grain moth (hereafter also referred to as ‘eggs’). After that, males and females were kept together for mating for 2 more days feeding on eggs. Then (6 days after emergence) randomly selected females were individually placed in Petri dishes (90 mm × 15 mm) lined with paper and distributed among the four experimental treatments which differed only in diet. All four diets included the following components: (1) the grain moth eggs glued to a piece of hard paper with sugar solution (the eggs were provided in excess; pilot tests showed that egg-laying females consume about 6 mg of eggs per day), (2) Eppendorf tube filled with water and plugged with a cotton ball, and (3) bean seedling. The difference between the diets was only in the number of aphids placed daily on the bean seedling: 0 (control), 2, 10, and about 50 (pilot tests showed that egg-laying females consumed about 40 aphids per day). In this experiment, only *M. persicae* older nymphs and adults were used, their average weight was about 0.3 mg. Each replicate of each treatment was started with 30–32 females. The experiment lasted until the 26th day after female emergence. In all treatments, female survival was checked, food was replaced, and the number of laid eggs was recorded daily.

In the course of the experiment dissections of randomly selected females were performed by the following scheme: 4 females per replicate were dissected on the first day after emergence and 4 females per replicate were dissected just before the distribution among treatments (i.e., 6 days after emergence). Then 2 females per each replicate of each treatment were dissected 7, 8, 9, 10, 11, 12, 14, 16, 18, 20, 22, and 24 days after emergence. At the end of the experiment (26 days after emergence) all survived females were dissected. Thus, the duration of the experiment (the ‘observation period’) for an individual female was from 1 to 20 days (the age at dissection, correspondingly, was from 7 to 26 days after emergence).

The state of the ovaries and fat body of dissected females was determined with the following scales similar to those used in our previous study [20] and in earlier studies by other authors [73,74,75,76,77,78] (Figure 1).

For ovaries:Not developed (oocytes are not visible; follicles are absent; no or hardly any constrictions of ovarioles);Poorly developed (follicles are few and small; ovarioles are somewhat constricted around the follicles);Moderately developed (follicles are of medium size and become vitellinized; ovarioles are markedly swollen; the largest oocyte is filled with whitish yolk);Fully developed (follicles are vitellinized; mature oocytes are present; the largest oocyte reached the maximum size).

In addition, the signs of resorption in the largest follicles (resorptive oocytes are smaller, their shape and structure are modified, and they are darker than normal oocytes) were recorded. Resorption was also recorded for all females who laid eggs but did have not fully developed ovaries upon dissection.

For fat body:Not developed (practically inconspicuous);Poorly developed (composed of small scattered globules interspersed around the internal organs);Moderately developed (internal organs are partially hidden by the fat body composed of well-formed globules);Fully developed (internal organs are completely hidden by the fat body composed of well-formed globules and interconnected stringy lobes).

For further analysis, all females were attributed to 4 reproductive states:Reproductively active (ovaries are fully developed).Intermediate (ovaries are moderately developed);Diapausing (ovaries are not or poorly developed; the fat body is moderately or fully developed);Underdeveloped (both ovaries and fat body are not or poorly developed).

Finally, the length and width of the largest follicle were measured for all reproductively active females (i.e., for those with fully developed ovaries), and its size was calculated using the following formula:S = 1/6 π L W^2^
where S—follicle size (volume), L—follicle length (corresponds to the axis of rotational symmetry), and W—follicle width (corresponds to the axis perpendicular to length) [79,80]. The measuring was performed using a stereomicroscope fitted with an ocular micrometer.

As noted above, each replicate of each of the four experimental treatments was started with 30–32 females. Although some females died during the experiment, at least 2 females per each replicate of each treatment were dissected at the end of each observation period. Six replicates of the experiment were performed and thus each observation period was represented by at least 12 females (in total, the main data set included 765 dissected females). For each female, the following parameters were recorded or calculated.

The duration of the pre-oviposition period (from adult emergence to the first oviposition, only for females that laid eggs);Total fecundity (the number of eggs laid during the observation period);Oviposition intensity (the mean number of eggs laid daily calculated for the period from the onset of egg-laying to the end of observation);The degree of ovaries development at dissection;The degree of fat body development at dissection;The reproductive state at dissection;The signs of resorption at dissection;The size of the largest follicle at dissection (for reproductively active females);The onset of egg-laying (yes or no).

All these parameters were analyzed in relation to the two experimental factors: diet (the number of aphids per day: 0, 2, 10, and 50) and the duration of the observation period (from 1 to 20 days which corresponded to female age at dissection from 7 to 26 days).

### 2.3. Statistical Analysis

For non-parametric data (reproductive state, the onset of egg-laying, the signs of resorption, the degrees of ovaries, and fat body development) the binary probit analysis and the chi-square test were used. Parametric data (the duration of the preoviposition period, total fecundity, egg-laying intensity, and follicle size) were ranked because of their non-normal distribution and analyzed by GLM. Multiple pairwise comparisons were made with Tukey’s HSD test of ranked data. In the text and figures, untransformed data are given. All calculations were conducted using SYSTAT 10.2 software (Systat Software Inc., Richmond, VA, USA) [81].

## 3. Results

### 3.1. Reproduction Parameters

Survival of *C. propinqua* females during the experiment was high (94.9–100% depending on diet) and marginally significantly increased with the number of aphids provided daily (Table 1). The proportion of females that started to lay eggs during the observation period significantly depended on the age at dissection (i.e., at the end of the observation period) and on the number of aphids consumed daily (Table 1, Figure 2a). As seen in Table 2, the percentage of egg-laying females significantly increased when the number of aphids provided daily increased from 0 to 2 and from 2 to 10 but the difference between the treatments with 10 and 50 aphids was not statistically significant. Pre-oviposition period, on the contrary, was not significantly dependent on diet (Table 1 and Table 2). Total fecundity (the number of eggs laid during the observation period) of egg-laying females depended both on the age at dissection and on diet (Table 1, Figure 2c). As well as the proportion of egg-laying females, their fecundity increased with the number of daily provided aphids but the pattern of dependence was quite different: fecundity significantly differed between the treatments with 2 and 10 and with 10 and 50 aphids but not between the treatments with 0 and 2 aphids. Moreover, in the treatment with two aphids the mean fecundity of egg-laying females was even (not significantly) lower than in the control (Table 2). Oviposition intensity (the mean number of eggs laid daily during the period from the onset of egg-laying to the end of the observation period) of egg-laying females showed the same pattern of increase with the number of aphids consumed daily but decreased with the observation period (Table 1 and Table 2, Figure 2d).

For practical purposes, however, fecundity parameters should be calculated for all females, not only for egg-laying ones. As could be expected based on the above results, both total fecundity and oviposition intensity calculated for all tested females also increased with the number of aphids provided daily (Table 1, Figure 2e,f). The pattern of these increases was similar to that of egg-laying individuals: the absence of difference between the treatments with zero and two aphids and a sharp growth with the further increase in the number of aphids consumed daily (Table 2).

### 3.2. Dissection Results

Dissections showed that at emergence both ovaries and fat bodies of all *C. propinqua* females were not or poorly developed. However, after 6 days of feeding on the grain moth eggs ovaries of most females were moderately (33%) or even fully (25%) developed. Fat body in 6 days old females was usually (83%) poorly developed; moderately developed fat body (17%) was observed only in some females with poorly developed ovaries. At this time females were distributed among treatments and then their reproductive state was markedly dependent on diet and on the age at dissection (Table 2 and Table 3). In particular, the proportion of reproductively active females increased with time and with the number of aphids consumed daily (Figure 3a, Table 2 and Table 3). Correspondingly, percentages of other reproductive states generally decreased both with time and with the number of aphids (Figure 3b–d; Table 2 and Table 3). It should be particularly noted that the incidence of diapause (as well as the proportion of egg-laying females) was substantially different in the treatments with 0 and 2 aphids whereas the difference between the treatments with 10 and 50 aphids was not statistically significant (Table 2). The incidence of resorption also decreased with the number of aphids provided daily but the pairwise differences between the treatments with 0 and 2 and with 10 and 50 aphids were not statistically significant whereas between the treatments with 2 and 10 aphids a kind of threshold was observed (Table 2 and Table 3). Finally, the size of the largest follicle depends only on diet (Table 3). As well as fecundity and oviposition intensity, follicle size significantly differs between the treatments with 2 and 50 aphids, but not between the treatments with 0 and 2 aphids (Table 2).

The degree of fat body development strongly depended on diet whereas the dependence on the age at dissection was not significant (Table 3). As well as the percentage of diapausing females, the mean degree of fat body development decreased with an increase in the number of aphids provided daily; in particular, the difference in the proportion of females with moderately or fully developed fat bodies between the treatments with 0 and 2 aphids per day was about 20% (Figure 4a, Table 2). Degrees of development of ovaries and fat body were strongly negatively correlated even in females fed on the same diet (Table 4). Therefore, the influence of diet on fat body development was evaluated separately for reproductively active (Figure 4b) and inactive (Figure 4c) females. The comparison of these two figures clearly illustrates fat body reduction in reproductively active females. In addition, it is seen that feeding on low-quality prey promotes fat accumulation even within the same category of females.

## 4. Discussion

First, we conclude that the fundamental aim of our study was achieved: the experiment allowed the separation of signal and nutritional effects. Indeed, although both the proportion of females that started to lay eggs (at least partly determined by the signal effect of food) and their egg-laying intensity (determined by the nutritional effect) increase with the number of aphids consumed daily, the patterns of these effects are quite different. In particular, the daily consumption of two aphids significantly increases the proportion of ovipositing females but has no impact on their mean fecundity (Table 2). Evidently, two aphids per day are enough to induce the onset of oogenesis in a substantial proportion of *C. propinqua* females, but the nutritional value of two aphids is too small to significantly increase the oogenesis intensity. Therefore, the threshold of the signal effect is lower than that of the nutritional effect.

Regarding earlier studies, proportional increase in fecundity with the daily consumption rate (i.e., the nutritional effect of diet) has been shown for numerous predatory ladybirds: *Coccinella trifasciata* Mulsant and *C. californica* Mannerheim [82], *Semiadalia undecimnotata* Schneider [83], *C. septempunctata* L. [84,85], *Harmonia axyridis* (Pallas) [18,69,74], *Chilocorus nigritus* F. [86], *C. transversoguttata richardsoni* Brown [85], *Scymnus subvillosus* (Goeze) [87], *Anegleis cardoni* (Weise) [70], *Adalia tetraspilota* (Hope) [71] and *Hippodamia variegata* (Goeze) [71]. Hodek and Evans [7] generalize that above a certain threshold (determined by the need for self-maintenance) fecundity increases linearly with food consumption.

The signal effect of diet (i.e., the influence on the hormonally mediated developmental switch between reproduction and adult diapause) has been rarely investigated. Zaslavskiy et al. [15] showed both the rate of reactivation from reproductive diapause and the mean fecundity of *Harmonia sedecimnotata* Fabr. females increased with the number of aphids consumed daily, but the patterns of these effects were markedly different. Moreover, the first stages of reactivation were induced by the smell of aphids, which directly proved the signal effect of food. Both nutritional and (supposedly) signal effects of the daily feeding rate were also shown for *H. axyridis* [18,88], *A. cardoni* [70], *A. tetraspilota, H. variegata* [71], and *Propylea japonica* (Thunberg) [39], although the authors of the last study suggested that the delayed maturation can be explained by that diet-restricted females needed a longer period to accumulate nutrients. For *H. axyridis* even the threshold of the response was determined: females kept with three aphids laid eggs whereas those kept with two aphids did not mature [69]; similar results were later obtained by Ovchinnikova et al. [72]. In *H. sedecimnotata* this threshold is also about two aphids/day although some females matured even when provided with one aphid daily [15]. Oviposition of *P. japonica* females feeding on an artificial diet can be also induced by adding two aphids per day [39]. Thus, our results agree with earlier studies conducted on different ladybird species.

It is known that the resorption of oocytes (oosorption) in synovigenic insects and, particularly, in ladybirds is an adaptive response to the absence or scarcity of food [22,74,89,90,91,92,93]. Resorption can be induced by 2–3 days of starvation [85] or even sooner [74]. Our experiment shows that the threshold of the induction of egg resorption in *C. propinqua* females is between 2 and 10 aphids per day. Rapid resorption during food scarcities and the immediate start of ovarian development under favorable conditions is a common adaptive strategy of predatory ladybirds [26,74,85]. In addition, resorption plays a role in balancing energy allocation between oocyte production and the search for suitable oviposition sites. Thus, the ovary can be used as a kind of energy storage system [21,22,74,92].

An increase in the largest follicle size with the number of aphids provided daily (Table 2 and Table 3, Figure 3f) can be most probably considered a consequence of an increase in oviposition intensity. Follicle size was measured only in reproductively active (usually egg-laying) females and thus the largest follicle can be small only in individuals that have recently laid an egg but the next follicle is not yet mature. Evidently, with a random selection of females, the probability of such an event increases with the time between ovipositions and hence decreases with an increase in oviposition intensity. It should be noted that the size of the largest follicle and oviposition intensity shows the same pattern of the trophic response: a small (not statistically significant) difference between the treatments with 0 and 2 aphids and sharp increase in the intervals from 2 to 10 and from 10 to 50 aphids/day (Table 2).

Accumulation of fat and other nutrients is an essential component of the diapause syndrome in insects [94,95,96] and, particularly, in ladybirds [13,17,20]. In our experiment, most of reproductively inactive females had a moderately or fully developed fat body, i.e., were in the state of reproductive diapause whereas in only a few ‘underdeveloped’ individuals both ovaries and fat bodies were not or poorly developed. This strong negative correlation between the development of ovaries and fat body can be considered as another proof that the long-term delay or termination of egg-laying by *C. propinqua* females fed on the restricted diet is not a simple direct physiological result of the deficit of nutrients but a hormonally mediated developmental switch from oogenesis to the induction of reproductive diapause. The same is suggested by the independence of the pre-oviposition period on diet: if the hormonal switch is triggered by an appropriate signal, the nutritional value of food influences oviposition intensity but not the onset of egg-laying as it is.

The applied conclusions of our study are also quite clear. The daily addition of a relatively small amount of high-quality prey (10 individuals of the green peach aphid per day) to low-quality food (the grain moth eggs) causes a substantial (more than twofold) increase in the egg production by the predatory ladybird population (Table 2). Further increase in the amount of high-quality food also results in a marked increase in fecundity, but this second doubling is less effective because it required a five-fold increase in the daily portion (50 aphids/day). Our sporadic observations showed that in this treatment ladybirds fed mostly (if not exclusively) on aphids. As was noted in the Introduction, various types of mixed diets were repeatedly tested with numerous predatory ladybirds [6,7,8,23]. In some cases, the fecundity of females feeding on mixed diets was higher than that of females feeding on each of their components. For example, the fecundity of *C. septempunctata* and C. *transversoguttata* females feeding on the mixture of a low-quality (larvae of the alfalfa weevil *Hypera postica* (Gyllenhal)) and a high-quality (pea aphids *Acyrthosiphon pisum* (Harris)) food was higher than that of females feeding on each of these prey [28]. *Harmonia axyridis* maturation rate and fecundity also increased when females were reared on the mixture of two aphid species [32], on the mixture of pollen with *Ephestia kuehniella* Zeller eggs [34], and on the mixture of the grain moth eggs with nymphs and adults of the green peach aphid [18]. The addition of supplementary plant food increased the fecundity of *Hippodamia convergens* Guerin–Meneville females feeding on their suitable aphid prey although females fed only on plant food did not lay eggs [40]. Similarly, females of *P. japonica* fed either on an artificial diet or on frozen aphids did not lay eggs, whereas those fed on the mixture of the two foods oviposited [39]. However, many other studies, on the contrary, showed that the quality of a mixed diet was not higher than the quality of its components [25,27,29,30,33,38,41]. The same is suggested by our results.

For the economic interpretation of the results of the present study, the following calculations can be made. The experiment showed that the average fecundity of *C. propinqua* females that consumed 0 (control), 10, and 50 aphids per day was about 45, 103, and 238 eggs/female, correspondingly (Table 2). Thus, the addition of 10 and of 50 aphids to the control diet increases fecundity by 103 − 45 = 58 and by 238 − 45 = 193 eggs/female that is by 129% and by 429% of the control fecundity, correspondingly. This addition would be economically feasible if an increase in the cost of food would be lower than an increase in fecundity. The control (basic) diet in our study included only eggs of the grain moth and each female consumed about 6 mg of eggs per day (see Section 2.2). To achieve the economic feasibility of the addition of 10 aphids per day the cost of 10 aphids should be lower than 129% of the cost of 6 mg of eggs (i.e., the price of one aphid should be lower than 129/10 = 12.9% of the price of 6 mg of eggs) whereas to achieve economic feasibility of the addition of 50 aphids per day the cost of 50 aphids should be lower than 429% of the cost of 6 mg of eggs (i.e., the price of one aphid should be lower than 429/50 = 8.6% of the price of 6 mg of eggs). We conclude that (1) the addition of 10 aphids (other conditions being the same) would be more economically feasible than the addition of 50 aphids and (2) the addition of 10 aphids would be feasible if the cost of one aphid would be lower than the cost of 6 × 0.129 = 0.77 mg of the grain moth eggs (i.e., the cost of 1000 aphids would be lower than the cost of 0.77 g of eggs). Considering that, for example, the current commercial price of 1 g of the grain moth eggs in St. Petersburg (Russia) is 73 rubles (ca 0.9 USD) [97] this means that the price of 1000 aphids should be less than 73 × 0.77 = 56 rubles (ca 0.7 USD). Unfortunately, we did not find any Russian commercial supplier of aphids, but a recent estimate [98] showed that the net cost of 1 g (ca 3000 individuals) of laboratory-reared green peach aphid is 339 rubles and, hence, 1000 aphids costs 339/3 = 113 rubles (ca 1.4 USD) that is 2 times higher than the above estimated upper threshold. Of course, these calculations are merely a very rough estimation and relative prices of different products are very variable. Besides, the net cost of mass production is much lower than that of laboratory rearing. However, it is clear that the cost-effective use of aphids as an addition to the grain moth eggs would require substantial improvement in rearing methods. In this regard, the use of an artificial diet for mass rearing of aphids seems particularly promising [99].

Regarding biological control of pests by preventing colonization [4,60,61,62,63,64,65,66,67,68], we conclude that when supplied with the grain moth eggs, *C. propinqua* females are able to survive for a long time under greenhouse conditions. In the absence of aphids, their oviposition intensity will be very low, but the appearance of pests will result in a corresponding increase in both the proportion of ovipositing females and their fecundity.

## Figures and Tables

**Figure 1 insects-14-00587-f001:**
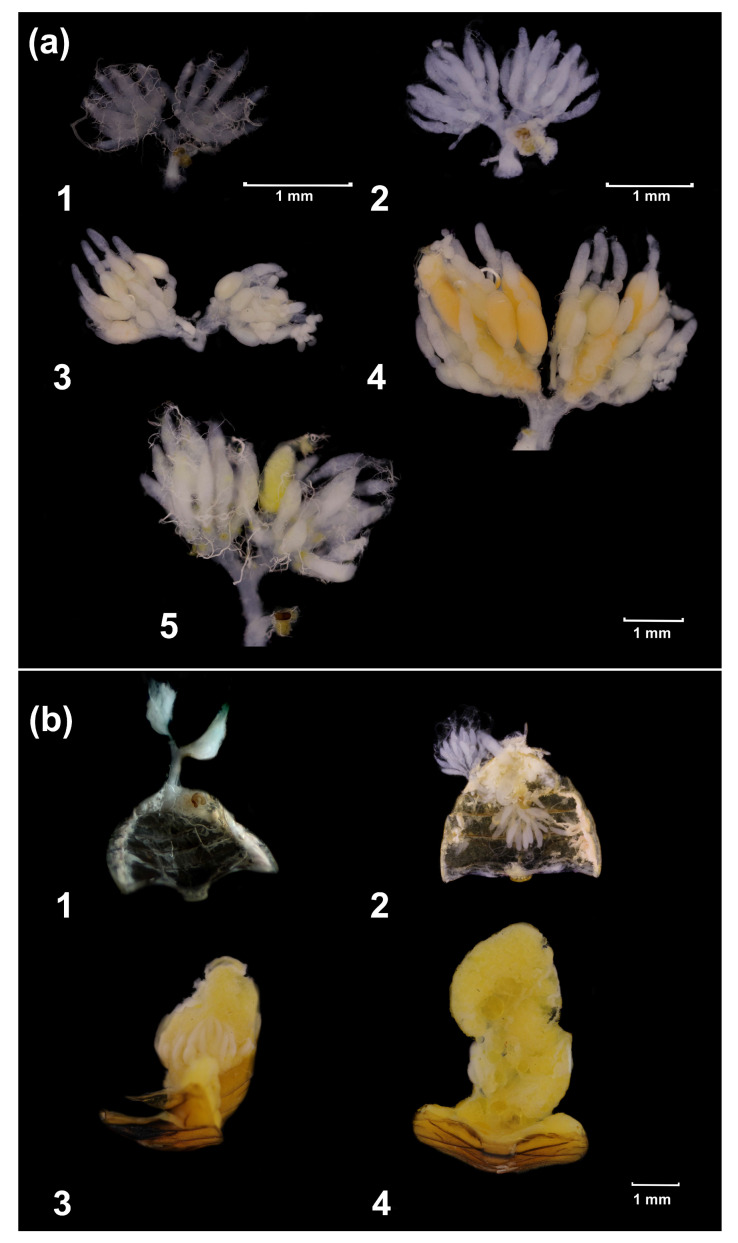
States of ovaries and the fat body in *Cheilomenes propinqua* females. (**a**)—ovaries: 1—not developed, 2—poorly developed, 3—moderately developed, 4—fully developed, 5—resorption; (**b**)—fat body: 1—not developed, 2—poorly developed, 3—moderately developed, 4—fully developed.

**Figure 2 insects-14-00587-f002:**
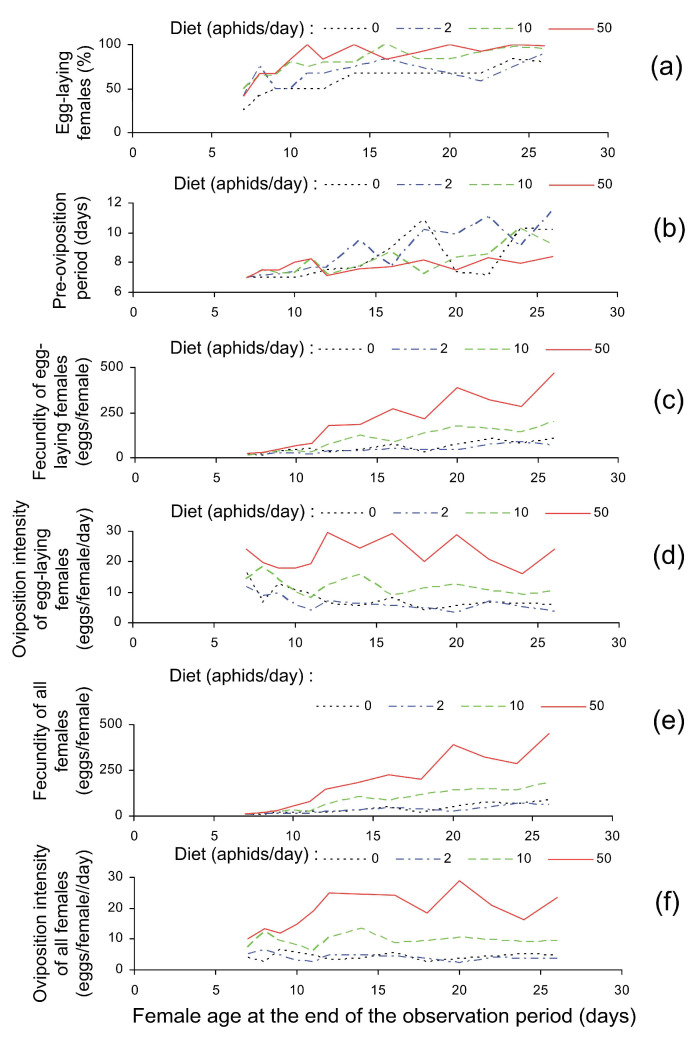
*Cheilomenes propinqua* reproduction parameters in relation to diet (the number of aphids offered daily) and female age at dissection (i.e., at the end of the observation period): (**a**)—the percentage of egg-laying females (females that laid at least one egg during the observation period), (**b**)—the mean pre-oviposition period (time from adult emergence to the first oviposition calculated only for females that laid eggs), (**c**)—the mean total fecundity calculated for egg-laying females (the number of eggs laid during the observation period calculated only for females that laid eggs), (**d**)—oviposition intensity calculated for egg-laying females (the mean number of eggs laid daily calculated for the period from the onset of egg-laying to the end of observation for females that laid eggs), (**e**)—the mean total fecundity calculated for all females (the number of eggs laid during the observation period; for females that did not lay eggs, it was taken to be zero), (**f**)—oviposition intensity calculated for all females (the mean number of eggs laid daily calculated for the period from the onset of egg-laying to the end of observation; for females that did not lay eggs it was taken to be zero). Some lines are slightly shifted to avoid overlap.

**Figure 3 insects-14-00587-f003:**
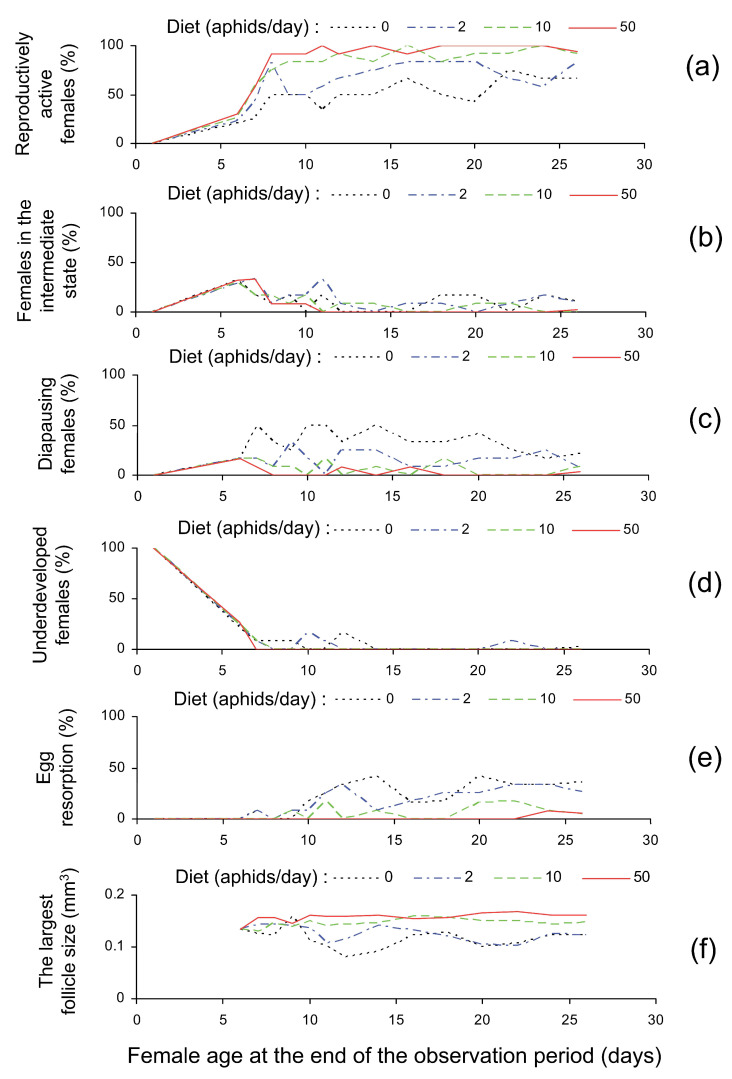
Reproductive state of *Cheilomenes propinqua* females at dissection in relation to diet (the number of aphids offered daily) and age (days from emergence): (**a**)—the percentage of reproductively active females (females with fully developed ovaries: follicles are vitellinized; mature oocytes are present; the largest oocyte reached the maximum size), (**b**)—the percentage of females in the intermediate state (females with moderately developed ovaries: follicles are of medium size and become vitellinized; ovarioles are markedly swollen; the largest oocyte is filled with whitish yolk), (**c**)—the percentage of diapausing females (ovaries are not or poorly developed; fat body is moderately or fully developed), (**d**)—the percentage of underdeveloped females (both ovaries and fat body are not or poorly developed), (**e**)—the percentage of females with egg resorption (resorptive oocytes are smaller, their shape and structure are modified, they are darker than normal oocytes; resorption was also recorded for all females which laid eggs but have not fully developed ovaries upon dissection), (**f**)—the size (volume) of the largest follicle (mm^3^). See Section 2.2 and Figure 1 for more explanations. Some lines are slightly shifted to avoid overlap.

**Figure 4 insects-14-00587-f004:**
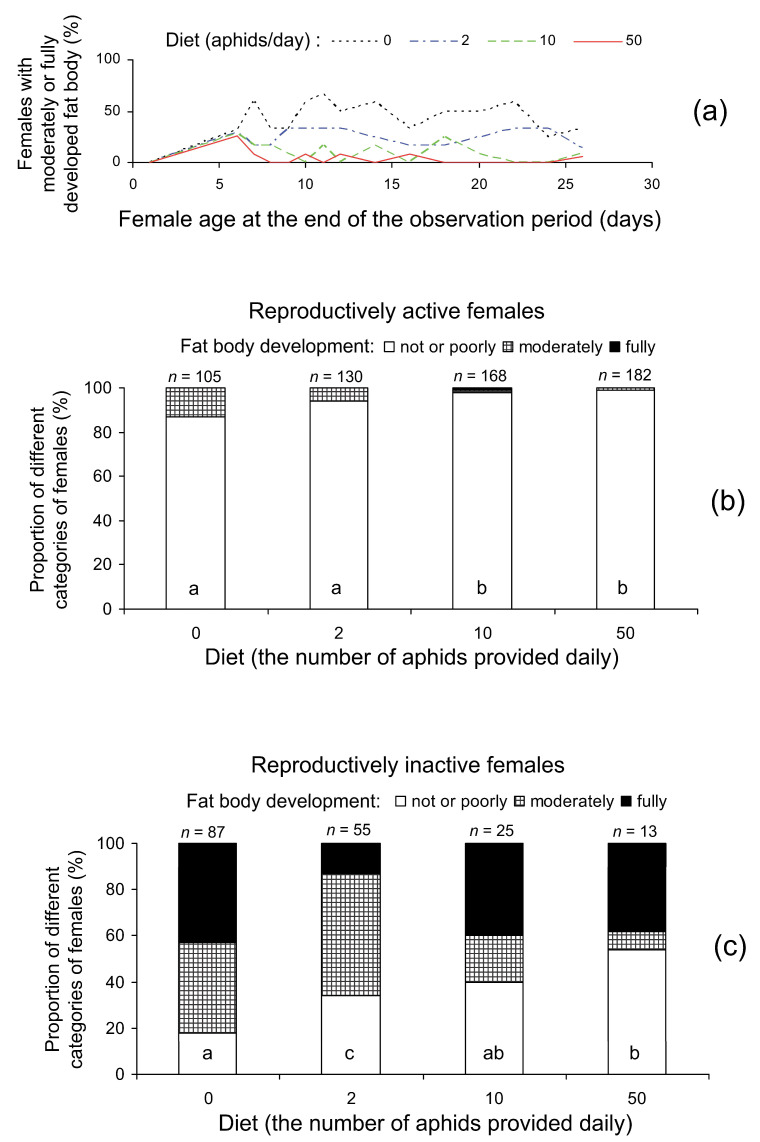
Fat body development in *Cheilomenes propinqua* females: (**a**)—the percentage of females with moderately or fully developed fat bodies at dissection in relation to diet (the number of aphids offered daily) and age (days from emergence), (**b**)—fat body development in reproductively active females in relation to diet, (**c**)—fat body development in reproductively inactive females in relation to diet. In (**b**,**c**): the percentage of females is shown; above the bars, sample size *n* is indicated. Different letters inside the bars indicate a significant difference (*p* < 0.05 by the chi-square test) between corresponding treatments. Some lines in (**a**) are slightly shifted to avoid overlap.

**Table 1 insects-14-00587-t001:** The significance of the influence of diet and duration of the observation period on reproduction parameters of *Cheilomenes propinqua* females (the regression coefficient *c* and the significance of influence *p*).

Parameters and Sample Size		Factors	
		Diet (the Number of Aphids per Day)	The Age at Dissection (Days)
Survival (%, *n* = 782) ^1^		*c* = 0.034, *p* = 0.076	– ^3^
The onset of egg-laying (yes or no, *n* = 765) ^1^		*c* = 0.016, *p* < 0.001	*c* = 0.062, *p* < 0.001
For females that laid eggs	Pre-oviposition period (days, *n* = 584) ^2^	*c* = −0.064, *p* = 0.321	*c* = 3.531, *p* < 0.001
	Total fecundity (eggs, *n* = 584) ^2^	*c* = 3.329, *p* < 0.001	*c* = 9.543, *p* < 0.001
	Egg-laying intensity (eggs/day, *n* = 584) ^2^	*c* = 4.190, *p* < 0.001	*c* = −2.722, *p* = 0.002
For all females	Total fecundity (eggs, *n* = 765) ^2^	*c* = 9.0489, *p* < 0.001	*c* = 13.247, *p* < 0.001
	Egg-laying intensity (eggs/day, *n* = 765) ^2^	*c* = 4.859, *p* < 0.001	*c* = 4.379, *p* < 0.001

^1^ The results of the binary probit analysis. ^2^ The results of the GLM analysis. ^3^ No data.

**Table 2 insects-14-00587-t002:** Various reproduction parameters of *Cheilomenes propinqua* females in relation to diet ^1^.

Reproduction Parameter	Diet (the Number of Aphids Offered Daily)			
	0	2	10	50
Egg-laying females (%)	58.6 ± 4.4 a	67.4 ± 4.0 b	81.43 ± 3.9 c	85.6 ± 4.9 c
The mean pre-oviposition period (days)	8.7 ± 0.3 a	9.4 ± 0.4 a	8.4 ± 0.2 a	7.9 ± 0.2 a
The mean total fecundity calculated for egg-laying females (eggs/female)	71.1 ± 7.2 a	47.5 ± 5.1 a	123.0 ± 9.1 b	271.1 ± 17.8 c
Oviposition intensity calculated for egg-laying females (eggs/female/day)	7.0 ± 0.6 a	5.5 ± 0.4 a	11.4 ± 0.6 b	22.9 ± 1.0 c
The mean total fecundity calculated for all females (eggs/female)	44.5 ± 5.1 a	33.9 ± 4.0 a	102.6 ± 8.3 b	237.8 ± 16.9 c
Oviposition intensity calculated for all females (eggs/female/day)	4.4 ± 0.4 a	3.9 ± 0.4 a	9.5 ± 0.6 b	20.1 ± 1.0 c
Reproductively active females (%)	51.9 ± 3.9 a	67.9 ± 4.2 b	85.9 ± 3.0 c	93.1 ± 3.1 d
Females in the intermediate state (%)	9.1 ± 2.2 ab	12.9. ± 2.9 a	7.1 ± 1.8 ab	4.6 ± 1.6 b
Diapausing females (%)	35.6 ± 3.3 a	15.9 ± 2.6 b	6.4 ± 1.9 c	2.2 ± 1.0 c
Underdeveloped females (%)	3.3 ± 1.5 a	3.2 ± 1.5 a	0.6 ± 0.6 a	0.0 ± 0.0 a
Females with egg resorption (%)	23.2 ± 4.1 a	19.3 ± 3.2 a	6.1 ± 1.9 b	1.1 ± 0.8 b
The size of the largest follicle (mm^3^)	0.117 ± 0.004 a	0.125 ± 0.003 a	0.147 ± 0.002 ab	0.160 ± 0.002 b
Females with moderately or fully developed fat body (%)	46.8 ± 3.7 a	25.5 ± 2.3 b	9.0 ± 2.4 c	3.0 ± 1.1 d

^1^ Means and SEM for females ages from 7 to 26 days are given. Values with different letters in the same row are significantly different (*p* < 0.05 by the Tukey HSD test; before the test, the data were ranked separately for each age and then pooled).

**Table 3 insects-14-00587-t003:** The significance of the influence of diet and duration of the observation period on the state of *Cheilomenes propinqua* females at dissection (the regression coefficient *c* and the significance of influence *p*).

Parameters		Factors	
		Diet (the Number of Aphids per Day)	The Age at Dissection (Days)
Reproductive state at dissection (*n* = 765)	Reproductively active ^1^	*c* = 0.026, *p* < 0.001	*c* = 0.038, *p* < 0.001
	Intermediate ^1^	*c* = −0.011, *p* = 0.007	*c* = −0.032, *p* = 0.001
	Diapausing ^1^	*c* = −0.028, *p* < 0.001	*c* = −0.020, *p* = 0.018
	Underdeveloped ^1^	*c* = −0.077, *p* = 0.059	*c* = −0.056, *p* = 0.011
	Resorption ^1^	*c* = −0.032, *p* < 0.001	*c* = 0.041, *p* < 0.001
Follicle size (mm^3^) ^2^		*c* = 3.522, *p* < 0.001	*c* = 0.597, *p* = 0.523
Fat body development (*n* = 765) ^1^		*c* = −0.031, *p* < 0.001	*c* = −0.014, *p* = 0.070

^1^ The results of the binary probit analysis. ^2^ The results of the GLM analysis.

**Table 4 insects-14-00587-t004:** Correlation between degrees of development of ovaries and the fat body in relation to diet.

Correlation between Degrees of Development of Ovaries and Fat Body	Diet (the Number of Aphids Provided Daily)			
	0	2	10	50
Spearman rank correlation coefficient, *ρ*	*ρ =* −0.763	*ρ =* −0.688	*ρ =* −0.731	*ρ =* −0.634
Chi-square χ^2^, sample size *n*, and significance *p*	χ^2^ = 148.4, *n* = 192, *p* < 0.001	χ^2^ = 110.5, *n* = 185, *p* < 0.001	χ^2^ = 175.7, *n* = 193, *p* < 0.001	χ^2^ = 188.7, *n* = 195, *p* < 0.001

## Data Availability

Data can be obtained upon request from the corresponding author.

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
