# Peer review of "Signal and Nutritional Effects of Mixed Diets on Reproduction of a Predatory Ladybird, *Cheilomenes propinqua"

_insects, 2023, doi:10.3390/insects14070587_

Round 1
Reviewer 1 Report
In order to study the effect of food on the incidence of diapause and the quantity of oviposition and to select a low cost diet for laboratory and mass rearing of Cheilomenes propinqua, the authors monitored the effect of low quality prey and high quality prey on oviposition of this species.
Experiments are appropriately chosen and carefully made. The results are of high quality, but they are presented in some places in a way that isdifficult to understand (see note on l 308-319)
English is well understood to me but may needed some stylistic corrections to make the text completely flawless. As a non-native speaker, I cannot provide the qualified help.
Specific comments
l. 139 provide geographic coordinates of sampling sites
l. 142 "Before the experiment" - indicate approximately how long
l. 148-150 I would suggest to move the sentence "The experiment was conducted at the same photothermal conditions that were shown to be close to optimal for reproduction of this species [20]" to the top of the paragraph
l. 155 "differed" replace with "which differed"
l. 208 here it would be possible to state for which geometric body the formula applies, S – size = volume?
l. 226 and 227 the meaning of the word "rate" is not entirely clear here
l. 252 "marginally not significantly" - delete "not"
l. 308 and 339 I recommend a more detailed description of the meaning of the figure directly in the figure captions, even if the explanation given in Methods is repeated.
l. 318-319 "the absence of letters means the absence of any significant pairwise difference." - perhaps it is better to mark all values with one letter - "a"
Discussion: well written
Author Response
Dear reviewer:
With many thanks, we have considered and accepted all your comments. Please, see our replies below.
l. 139 provide geographic coordinates of sampling sites
>> Geographic coordinates are given (line 139).
l. 142 "Before the experiment" - indicate approximately how long
>> The ladybirds were reared in Zoological Institute RAS for about one year (line 142).
. 148-150 I would suggest to move the sentence "The experiment was conducted at the same photothermal conditions that were shown to be close to optimal for reproduction of this species [20]" to the top of the paragraph
>> This sentence was moved (lines 146-147)
l. 155 "differed" replace with "which differed"
>> Corrected (line 156)
l. 208 here it would be possible to state for which geometric body the formula applies, S – size = volume?
>> Yes, S is volume which is indicated in the revised manuscript (line 209)
l. 226 and 227 the meaning of the word "rate" is not entirely clear here
>> We are sorry for the unclear English. Yes, in fact, we mean “the degree of development”. This error is corrected here and everywhere in the manuscript (lines 227, 228, 334, 336 etc.), .
l. 252 "marginally not significantly" - delete "not"
>> Corrected (line 254)
l. 308 and 339 I recommend a more detailed description of the meaning of the figure directly in the figure captions, even if the explanation given in Methods is repeated.
>> More detailed explanations of the variables are given in the legends to the figures 2 and 3 (lines 310-322 and 348-360)
l. 318-319 "the absence of letters means the absence of any significant pairwise difference." - perhaps it is better to mark all values with one letter - "a"
>> Corrected (see Table 2).
Reviewer 2 Report
Dear Editor
This a very interesting manuscript about the improving of the fecundity and longevity of the predator Cheilomenes propinqua with the use of different mix of diets in order to apply in the mass rearing. The results reinforce the idea that there is a proposal which could be important to optimize the fecundity of predator.
In my opinion the presentation is excellent, a complete introduction, also the other sections as material and methods, adequate analysis of data and a very good discussion of the data.
My recommendation is to publish this manuscript with minor changes.
I have only a few comments:
L 330 What does mean “the difference……… was very large”?
L 477 prices of different goods (????) are very variable
Table 1, Table 3, and subtitle 3.1 (demographic parameters), I suggest to be more specific. Could be better to use fecundity or reproduction.
The english language in my opinion is enough to describe the most important data and their analysis.
Author Response
Dear reviewer:
With many thanks, we have considered and accepted all your comments. Please, see our replies below.
L 330 What does mean “the difference……… was very large”?
>> Exact difference (about 20%) is indicated (line 339)
L 477 prices of different goods (????) are very variable
>> We apologize for unclear English. The term “goods” was replaced by “products”. (line 494).
Table 1, Table 3, and subtitle 3.1 (demographic parameters), I suggest to be more specific. Could be better to use fecundity or reproduction.
>> The term “reproduction parameters” is now used both in the subtitle and in tables (lines 252, 282, 324).